# Differential Privacy Over Riemannian Manifolds

**Matthew Reimherr**
Department of Statistics
Pennsylvania State University
University Park, PA
mreimherr@psu.edu

**Karthik Bharath**
School of Mathematical Sciences
University of Nottingham
Nottingham, UK
Karthik.Bharath@nottingham.ac.uk

**Carlos Soto**
Department of Statistics
Pennsylvania State University
University Park, PA
cjs7363@psu.edu

## Abstract

In this work we consider the problem of releasing a differentially private statistical summary that resides on a Riemannian manifold. We present an extension of the Laplace or K-norm mechanism that utilizes intrinsic distances and volumes on the manifold. We also consider in detail the specific case where the summary is the Fréchet mean of data residing on a manifold. We demonstrate that our mechanism is rate optimal and depends only on the dimension of the manifold, not on the dimension of any ambient space, while also showing how ignoring the manifold structure can decrease the utility of the sanitized summary. We illustrate our framework in two examples of particular interest in statistics: the space of symmetric positive definite matrices, which is used for covariance matrices, and the sphere, which can be used as a space for modeling discrete distributions.

## 1 Introduction

Over the last decade we have seen a tremendous push for the development and application of methods in data privacy. This surge has been fueled by the production of large sophisticated datasets alongside increasingly complex data gathering technologies. One theme that has emerged with the proliferation of highly structured and dynamic data is the importance of exploiting underlying structures in the data or models to maximize utility while controlling disclosure risks. In this paper we consider the problem of achieving pure *Differential Privacy*, DP, when the statistical summary to be released takes values on a complete Riemannian manifold.

Riemannian manifolds are used extensively in the analysis of data or parameters that are inherently nonlinear, meaning, either addition or scalar multiplication may cause the summary to leave the manifold, or such operations are not even well defined. Classic examples of such objects include spatio-temporal processes, covariance matrices, projections, rotations, compositional data, densities, and shapes. Traditional privacy approaches for handling such objects typically consist of utilizing an ambient or embedding space that is linear so that standard DP tools can be employed. For example, Karwa and Slavković [2016] considered the problem of releasing private degree sequences of a graph which required them to project back onto a particular convex hull as a post-processing step. Such an approach is a natural starting point and reasonable so long as the space doesn't exhibit too much curvature. However, there are several interrelated motivations for working with the manifolds directly. First, if one employs an ambient space (known as taking an extrinsic approach), then calculations such as the sensitivity may depend on the dimension of the ambient space, which will in turn impact

35th Conference on Neural Information Processing Systems (NeurIPS 2021).

the utility of the private statistical summary. For example, minimax rates in DP typically scale polynomially in the dimension [e.g. Hardt and Talwar, 2010, Bun et al., 2018, Kamath et al., 2019]. Second, the Whitney embedding theorem states that, in the worst case, to embed a manifold in Euclidean space requires a space that is twice the dimension of the manifold. Third, if the manifold exhibits substantial curvature, then even small distances in the ambient space may result in very large distances on the manifold. Lastly, the choice of the ambient space may be arbitrary and one would ideally prefer if this choice did not play a role in the resulting statistical analysis.

**Related Literature:** To the best of our knowledge, general manifolds have not been considered before in the DP literature. The closest works come from the literature on private covariance matrix estimation and principal components [Blum et al., 2005, Awan et al., 2019, Amin et al., 2019, Kamath et al., 2019, Biswas et al., 2020, Wang and Xu, 2020]. While not always explicitly described in some works [Wang et al., 2013, Wei et al., 2016], these objects lie in nonlinear manifolds, namely, the space of symmetric positive definite matrices (SPDM) and the space of projections matrices respectively, called the Stiefel manifold. For example, in Chaudhuri et al. [2013] they consider the problem of generating a synthetic PCA projection by using the matrix Bingham distribution, a distribution over the Stiefel manifold [Khatri and Mardia, 1977, Hoff, 2009]. In contrast, producing private covariance matrix estimates usually involves adding noise in a way that preserves symmetry, but does not use any deeper underlying manifold structure. A related problem comes from the literature on private manifold learning [Choromanska et al., 2016, Vepakomma et al., 2021], though this is entirely distinct from the present work, which assumes the underlying manifold is known, usually because of some physical constraints on the data or statistical summaries.

**Contributions:** In this paper we utilize tools from Differential Geometry that allow us to extend the Laplace mechanism for $\epsilon$-*Differential Privacy* to general Riemannian manifolds. Under this framework, we consider the problem of privately estimating the Fréchet mean of data lying on a $d$-dimensional manifold. We are able to bound the global sensitivity of the mean and provide bounds on the magnitude of the privacy noise, as measured using the distance on the manifold, that match the optimal rates derived in Euclidean spaces. However, we demonstrate the influence of curvature of the space in understanding the sensitivity of the mean, and how the situation becomes especially challenging on positively curved spaces. We conclude by providing two specific numerical examples that elucidate this phenomenon: the first considers data coming from the space of positive definite matrices equipped with a geometry that results in negative curvature, while the second example considers data lying on the sphere, which has constant positive curvature.

## 2 Notation and Background

In this section we provide the basic notation, terminology, and mathematical concepts needed from differential geometry. For a more detailed treatment of differential geometry and Shape Analysis there are many excellent classic texts, e.g., Gallot et al. [1990], Lang [2002], Dryden [2014], Srivastava and Klassen [2016], Lee [2018], while an overview of DP can be found in Dwork and Roth [2014].

Throughout the paper we let $\mathcal{M}$ denote a $d$-dimensional complete Riemannian manifold. For $m \in \mathcal{M}$, denote the corresponding tangent space as $T_m\mathcal{M}$. We assume $\mathcal{M}$ is equipped with a Riemannian metric $\{\langle \cdot, \cdot \rangle_m : m \in \mathcal{M}\}$, which is a collection of inner products over the tangent spaces $\{T_m\mathcal{M} : m \in \mathcal{M}\}$ that vary smoothly in $m$.

Two quantities that will be used extensively in this work are that of the distance and volume induced from the Riemannian metric. Consider two points $m_1, m_2 \in \mathcal{M}$ and a smooth path $\gamma : [0,1] \to \mathcal{M}$ such that $\gamma(0) = m_1$ and $\gamma(1) = m_2$. The derivative $\dot{\gamma}(t)$ represents the velocity of $\gamma$ as it passes through the point $\gamma(t)$ and can thus be identified as an element of the tangent space $T_{\gamma(t)}\mathcal{M}$. We define the length of the curve as

$$L(\gamma) := \int_0^1 \langle \dot{\gamma}(t), \dot{\gamma}(t) \rangle_{\gamma(t)}^{1/2} \, \mathrm{d}t.$$

The distance between $m_1$ and $m_2$ is the infimum over all possible paths connecting the two points

$$\rho(m_1, m_2) := \inf_{\substack{\gamma : \gamma(0) = m_1 \\ \gamma(1) = m_2}} L(\gamma).$$

If this distance is achieved by a particular path, $\gamma$, then we say that $\gamma$ is a *geodesic*. Geodesics generalize the concept of a straight line to nonlinear spaces, and are thus a natural tool to consider

when generalizing DP perturbation mechanisms. The distance $\rho(\cdot, \cdot)$ defines a valid distance metric over $\mathcal{M}$; we say that $\mathcal{M}$ is complete if it is complete as a metric space. By the *Hopf-Rinow theorem* [Lee, 2018, Theorem 6.19], this is equivalent to saying that to every pair of points there exists a minimizing geodesic, though if the points are far enough apart it need not be unique.

In the next section we will use the *exponential map*, $\exp_m : T_m\mathcal{M} \to \mathcal{M}$, which is a means of moving between the manifold and tangent spaces. If there exists a unique geodesic, $\gamma$, between between points $\gamma(0) = m_1$ and $\gamma(1) = m_2$, then the exponential map is defined as the mapping $\exp_{m_1}(\dot{\gamma}(0)) = m_2$. In other words, $\exp_m$ maps initial velocities to points on $\mathcal{M}$ through the use of geodesic curves. If the manifold is complete, then the exponential map is locally diffeomorphic, meaning that for any $m \in \mathcal{M}$ there exists an open neighborhood $U_r(m)$ of $\mathcal{M}$ that is diffeomorphic to an open ball $B_r(\mathbf{0})$ centred at the origin of $T_m\mathcal{M}$. This ensures that the inverse $\exp_m^{-1} : U_r \to B_r(\mathbf{0})$ known as the inverse-exponential (also known as logarithm) map exists locally. The *injectivity radius of $m$* is the supremum of $r \mapsto U_r$ over all such radii $r$. The *injectivity radius of $\mathcal{M}$*, denoted by inj$\mathcal{M}$, is defined to be the infimum of the injectivity radii of *all* points $m \in \mathcal{M}$.

The Riemannian metric can be used to define a notion of volume, called the Riemannian volume measure denoted as $\mu$, which acts analogously to Lesbesgue measure in Euclidean space. To define the measure, it helps to employ a chart, $(U, \phi)$, although the final definition will not depend on which chart we choose. Since $\phi : U \to \phi(U) \subset \mathbb{R}^d$ is a homeomorphism, at each $m \in U$ the inverse $\phi^{-1}$ induces a basis, $\partial x_1, \ldots, \partial x_d$ on $T_m\mathcal{M}$ and a corresponding dual basis $dx^1, \ldots, dx^d$ on $T_m\mathcal{M}^*$, the dual space of $T_m\mathcal{M}$. Then the Riemannian volume form is defined as $\sqrt{|g|}dx^1 \wedge \cdots \wedge dx^d$, where $g_{ij} = \langle \partial x_i, \partial x_j \rangle_m$ and $|\cdot|$ is the absolute value of the determinant, which can be shown to be invariant to the choice of chart. This induces a volume over the set $U$ and upon employing a partition of unity, one can define $\mu$ over the entire manifold $\mathcal{M}$, equipped with the Borel $\sigma$-algebra.

## 3  Manifold Perturbations and Differential Privacy

Denote the dataset as $D = \{x_1, \ldots, x_n\}$ with the data coming from points $x_i$ collected from an arbitrary set $\mathcal{X}$. We aim to release a statistical summary $f(D)$ which takes values on $\mathcal{M}$. Defining differential privacy over a Riemannian manifold presents no major challenge since it is a well defined concept over any measurable space [Wasserman and Zhou, 2010, Awan et al., 2019], which includes Riemannian manifolds equipped with the Borel $\sigma$-algebra. Denote the (random) sanitized version of $f(D)$ as $\widetilde{f}(D)$. We can then define what it means for $\widetilde{f}(D)$ to satisfy $\epsilon$-DP.

**Definition 1.** *A family of randomized summaries, $\{\widetilde{f}(D) \in \mathcal{M} : D \in \mathcal{X}^n\}$, is said to be $\epsilon$-differentially private with $\epsilon > 0$, if for any adjacent database $D'$, denoted as $D \sim D'$, differing in only one record we have*

$$P(\widetilde{f}(D) \in A) \leq e^\epsilon P(\widetilde{f}(D') \in A),$$

*for any measurable set $A$.*

In a similar fashion, we can extend the notion of sensitivity to Riemannian manifolds using the distance function. However, this definition is by no means canonical and intimately connected to the type of noise one intends to use [Mirshani et al., 2019].

**Definition 2.** *A summary $f$ is said to have a global sensitivity of $\Delta < \infty$, with respect to $\rho(\cdot, \cdot)$, if for any two adjacent databases $D$ and $D'$ we have*

$$\rho(f(D), f(D')) \leq \Delta.$$

With a bounded sensitivity, it still isn't obvious how to produce a differentially private summary. In particular, we no longer have linear perturbations or classic noise mechanisms such as Laplace or Gaussian. However, the Riemannian structure allows us to use the volume measure as a base measure, similar to the Lebesgue measure on Euclidean spaces. We can then define a new distribution that can be viewed as a generalization of the Laplace distribution over manifolds [e.g. Hajri et al., 2016, for SPDM]. It is worth noting that this distribution is not equivalent to the multivariate Laplace in Euclidean settings, instead it is an instantiation of the K-norm distribution [Hardt and Talwar, 2010].

**Definition 3.** *A probability measure $P$ over $\mathcal{M}$ is called a Laplace distribution with footpoint $\eta \in \mathcal{M}$ and rate $\sigma > 0$ if for any measurable set $A$ we have*

$$P(A) = \int_A C_{\eta,\sigma}^{-1} e^{-\rho(\eta,m)/\sigma} \, \mathrm{d}\mu(m),$$

*where $0 < C_{\eta,\sigma} < \infty$ is the normalizing constant and $\mu$ is the Riemannian volume measure over $\mathcal{M}$.*

Unlike with Euclidean distance, the normalizing constant might only be finite for certain values of $\sigma$. The foot-point represents the center of the distribution but, in general, need not be equal to the mean of the distribution (which also might not exist). For data privacy, one often has to restrict the data or parameter space anyway, in which case the Laplace distribution can be restricted so that the normalizing constant, mean, etc are all finite and well defined. The advantage in using such a mechanism is that sensitivity can be readily transferred into differential privacy.

**Theorem 1.** *Let $f : \mathcal{X}^n \to \mathcal{M}$ be a summary with global sensitivity $\Delta$. Then the Laplace mechanism with footpoint $f(D)$ and rate $\sigma = 2\Delta/\epsilon$ satisfies $\epsilon$-differential privacy. If the normalizing constant, $C_{\eta,\sigma}$ does not depend on the footpoint, $\eta$, then one can take $\sigma = \Delta/\epsilon$.*

*Proof.* (*Sketch*) The proof follows from a direct verification via the triangle inequality. $\square$

In principle, it is possible to sample from the Laplace distribution using Markov Chain Monte Carlo (MCMC). One can start the chain at $\eta$ and then make small proposed steps by randomly selecting a direction and radius on the tangent space. The resulting tangent vector can be pushed to an element of $\mathcal{M}$ via the exponential map. Alternatively, if enough structure is known, as we illustrate in Section 5, one may be able to sample from the distribution directly or if the space is bounded then one can use rejection sampling.

An interesting alternative to our approach that is still inherently intrinsic is to instead generate the sanitised summary on a particular tangent space and then map it to the manifold using the exponential map. On the surface, this seems like a reasonable idea, however there are some subtle technicalities that would have to be overcome. In particular, one has to choose which tangent space to work with. Ideally one would work with the tangent space at the summary of interest, but that isn't private. If another plane is used, then there is the chance for more serious distortions from the noise. Likely these issues could be overcome, but would require additional work.

## 4   Differentially Private Fréchet Means

As before, suppose the data consists of $D = \{x_1, \ldots, x_n\}$, but now with $x_i \in \mathcal{M}$. The sample Fréchet mean, $\bar{x}$, is defined to be the global minimizer of the energy functional

$$\mathcal{M} \ni x \mapsto F_2(x) := \frac{1}{2n} \sum_{i=1}^{n} \rho^2(x, x_i),$$

which is a natural generalization of the Euclidean mean to manifolds. Conditions that ensure existence and uniqueness of $\bar{x}$ have been extensively studied since its inception in the 1970s [Karcher, 1977, Kendall, 1990]. Even when the mean is unique, the following example shows that the sensitivity need not decrease with the sample size, which produces sanitized estimates with low utility.

**Example 1.** Let $x_1, \ldots, x_n$ be points on the unit circle $\mathcal{M} = \mathcal{S}^1 = \{x \in \mathbb{R}^2 : \|x\| = 1\}$ with arc-length distance $\rho$, represented as angles such that $x_i = \frac{2\pi i}{n-1}, i = 1, \ldots, n-1$ and $x_n = x_{i'}$ for some $i' \in [n-1]$. Then minimizing $F_2$ occurs when we take $\bar{x} = x_i$. So, we can make the mean any of the $x_i$ by shifting a single point. If $n$ is even, the furthest any two points can be is $\pi$ and the resulting sensitivity is $\pi$, which clearly does not decrease with $n$.

The above example illustrates that one must have some additional structure to ensure that the sample Fréchet mean as a statistic is stable and that the sensitivity is properly decreasing with the sample size. The first requirement is that the sample Fréchet mean is unique; this imposes strong constraints on the spread of the data on $\mathcal{M}$ given by its curvature. Denote by $B_r(m)$ the open geodesic ball at $m$ of radius $r$ in $\mathcal{M}$. For a given dataset $D$ with points in $\mathcal{M}$ we make the following assumption.

**Assumption 1.** *The data $D \subseteq B_r(m_0)$ for some $m_0$, where $r < r^* := \frac{1}{2} \min\{\mathrm{inj}\mathcal{M}, \frac{\pi}{2}\kappa^{-1/2}\}$ and $\kappa > 0$ is an upper bound on the sectional curvatures of $\mathcal{M}$.*

For flat and negatively curved manifolds, Assumption 1 only says that the data lies in some bounded ball. In that case $\kappa \leq 0$ and we can interpret $\kappa^{-1/2}$ to be $+\infty$. Furthermore, the $\mathrm{inj}\mathcal{M}$ can be arbitrarily large. Thus the radius of this ball only impacts the sensitivity, which is very common in

data privacy. However, it is, in general, difficult to relax Assumption 1 for positively curved manifolds even when privacy isn't a concern. For example, it suffices to use the slightly weaker assumption $r < \frac{1}{2}\min\{\mathrm{inj}\mathcal{M}, \pi\kappa^{-1/2}\}$ to ensure that (i) the closure $\bar{B}_r(m_0)$ is geodesically convex; (ii) $\bar{x}$ exists and is unique; and (iii) $\bar{x}$ belongs to the closure of the convex hull of points in $D$ [Afsari, 2011]. For the unit sphere $\mathcal{S}^{d-1}$ in Example 1 we have $\mathrm{inj}\mathcal{M} = \pi$ and $\kappa = 1$, we require $r < \pi/2$ so that a dataset $D$ lying within a hemisphere on $S^{d-1}$ will have a unique sample Fréchet mean only if $D$ contains no point lying on the equator. However, we need the stronger Assumption 1 to ensure that $(x, y) \mapsto \rho^2(x, y)$ is convex along geodesics for *every* $x$ and $y$ inside $B_r(m_0)$ [Le, 2001], which is required to determine the sensitivity of $\bar{x}$.

In Theorem 2 we provide a bound on the global sensitivity of the Fréchet mean. The bound depends on the sample size $n$, the radius $r$ of the ball that contains the data, and a function $h(r, \kappa)$ which depends only on $r$ and on the upper bound $\kappa$ of the sectional curvatures of $\mathcal{M}$. For flat or negatively curved manifolds, we will see that $h(r, \kappa) = 1$, which matches classical results for the Euclidean space, owing to the classical Hadamard-Cartan theorem that states that a simply connected $\mathcal{M}$ with non-negative sectional curvatures is diffeomorphic to $\mathbb{R}^d$. However, the situation is more subtle for positively curved manifolds where $h$ can no longer be ignored.

**Theorem 2.** *Under Assumption 1 consider two datasets $D = \{x_1, \ldots, x_{n-1}, x_n\}$ and $D' = \{x_1, \ldots, x_{n-1}, x'_n\}$ differing by only one element. If $\bar{x}$ and $\bar{x}'$ are the two sample Fréchet means of $D$ and $D'$ respectively, then*

$$\rho(\bar{x}, \bar{x}') \leq \frac{2r(2 - h(r, \kappa))}{nh(r, \kappa)}, \qquad h(r, \kappa) = \begin{cases} 2r\sqrt{\kappa}\cot(\sqrt{\kappa}2r) & \kappa > 0; \\ 1 & \kappa \leq 0 \end{cases}.$$

*Proof.* Consider the energy functionals $F_2$ and $\widetilde{F}_2$ for the datasets $D$ and $D'$ with unique sample means $\bar{x}$ and $\bar{x}'$, respectively. Since $\mathcal{M}$ is complete the exponential map $\exp_x : T_x\mathcal{M} \to \mathcal{M}$ is surjective and under Assumption 1 the log map or inverse exponential map $\mathcal{M} \ni y \mapsto \exp_x^{-1}(y) \in T_x\mathcal{M}$ is well-defined for every $x \in B_r(m_0)$.

Here $x \mapsto \rho^2(x, y)$ is twice continuously differentiable, and under Assumption 1 the function $x \mapsto \rho(x, y)$ is strictly convex for all $x, y \in B_r(m_0)$ [Karcher, 1977, Afsari, 2011]. Consider an arc length parameterized, unit speed minimizing geodesic $\gamma$ between $\bar{x}$ and $\bar{x}'$ such that $\gamma(0) = \bar{x}, \gamma(b) = \bar{x}'$ with $b = \rho(\bar{x}, \bar{x}')$. The composition, $G_2 := F_2 \circ \gamma : [0, b] \to \mathbb{R}$ is now a twice continuously differential real-valued function with derivatives $\dot{G}_2$ and $\ddot{G}_2$, and thus

$$\dot{G}_2(b) = \dot{G}_2(0) + b\ddot{G}_2(t_0) = \rho(\bar{x}, \bar{x}')\ddot{G}_2(t_0),$$

for some $0 \leq t_0 \leq b$ since $\dot{G}_2(0) = 0$.

To determine $\ddot{G}_2(t_0)$, we need to calculate the second derivative of $\rho(\gamma(t_0 + \epsilon), q)^2$ evaluated at $\epsilon = 0$ and for an arbitrary $q \in B_r(m_0)$, which equals

$$2\left(\frac{\mathrm{d}}{\mathrm{d}\epsilon}\rho(\gamma(t_0 + \epsilon), q)\big|_{\epsilon=0}\right)^2 + 2\rho(\gamma(t_0), q)\frac{\mathrm{d}^2}{\mathrm{d}\epsilon^2}\rho(\gamma(t_0 + \epsilon), q)\big|_{\epsilon=0}.$$

Let $\beta_q$ be the angle between $\dot{\gamma}(t_0)$ and $\dot{\alpha}_q(t_0)$ formed in $T_z\mathcal{M}$, where $\alpha_q$ is a minimizing geodesic from $q$ to $\gamma(t_0)$; this implies that $\frac{\mathrm{d}}{\mathrm{d}\epsilon}\rho(\gamma(t_0 + \epsilon), q)\big|_{\epsilon=0} = \langle \nabla\rho(\gamma(t_0), q), \dot{\gamma}(t_0)\rangle_z = \cos\beta_q$, with $\nabla$ as the Riemannian gradient, since $\frac{\mathrm{d}}{\mathrm{d}t}\rho(\gamma(t), q)|_{t=0} = \dot{\gamma}(\rho(\bar{x}, q))$ and $\gamma$ is a unit-speed geodesic. As a consequence, with minimizing geodesics $\alpha_{x_i}$ from $x_i$ to $z = \gamma(t_0)$ and corresponding angles $\beta_{x_i}$,

$$\ddot{G}_2(t_0) = \frac{\mathrm{d}^2}{\mathrm{d}\epsilon^2}F_2(\gamma(t_0 + \epsilon))\big|_{\epsilon=0} = \frac{1}{n}\sum_{i=1}^{n}\left[\cos^2\beta_{x_i} + \rho(z, x_i)\frac{\mathrm{d}^2}{\mathrm{d}\epsilon^2}\rho(\gamma(t_0 + \epsilon), x_i)\big|_{\epsilon=0}\right].$$

Note that if $z = x_i$ for any $i$, the angle $\beta_{x_i}$ is not well-defined, but regardless of the chosen path $\alpha_{x_i}$ the contribution to the sum from the particular $x_i$ will be one. The Hessian $\frac{\mathrm{d}^2}{\mathrm{d}\epsilon^2}\rho(\gamma(t_0 + \epsilon), x_i)\big|_{\epsilon=0}$ of the distance function can be lower bounded using the Hessian comparison theorem [e.g. Lee, 2018, Theorem 11.7] to obtain

$$\ddot{G}_2(t_0) \geq \frac{1}{n}\sum_{i=1}^{n}[\cos^2\beta_{x_i} + a(\rho(z, x_i), \kappa)\sin^2\beta_{x_i}],$$

where

$$a(s,\kappa) = \begin{cases} s\sqrt{\kappa}\cot(\sqrt{\kappa}s) & \kappa > 0; \\ s^{-1} & \kappa = 0; \\ s\sqrt{|\kappa|}\coth(\sqrt{|\kappa|}s) & \kappa < 0. \end{cases}$$

If $\kappa > 0$, then $(s,\kappa) \mapsto a(s,\kappa) \leq 1$ and decreasing; on the other hand if $\kappa < 0$, $(s,\kappa) \mapsto a(s,\kappa) \geq 1$ and increasing. We hence have that

$$\ddot{G}_2(t_0) \geq h(r,\kappa) := \begin{cases} 2r\sqrt{\kappa}\cot(\sqrt{\kappa}2r) & \kappa > 0; \\ 1 & \kappa \leq 0, \end{cases} \tag{1}$$

since for $\kappa = 0$, $a(s,\kappa) \geq (2r)^{-1}$ and we can choose $r = 1/2$ since it is effectively unconstrained in this setting. The lower bound on $\ddot{G}_2(t_0)$ thus depends on whether $\mathcal{M}$ is positively or non-negatively curved depending on the sign of $\kappa$. This results in

$$\rho(\bar{x}, \bar{x}') \leq \frac{\dot{G}_2(b)}{h(r,\kappa)} = \frac{1}{h(r,\kappa)}[\dot{G}_2(b) - \dot{\tilde{G}}_2(b)],$$

where $\dot{\tilde{G}}_2(b) = 0$ since $\nabla\widetilde{F}_2(\bar{x}') = \mathbf{0}$. For any $x \in \mathcal{M}$, in normal coordinates, the gradients

$$\nabla\widetilde{F}_2(x) = -\frac{1}{n}\left[\sum_{i=1}^{n-1}\exp_x^{-1}(x_i) + \exp_x^{-1}(x_n')\right], \quad \nabla F_2(x) = -\frac{1}{n}\left[\sum_{i=1}^{n-1}\exp_x^{-1}(x_i) + \exp_x^{-1}(x_n)\right],$$

belong to $T_x\mathcal{M}$. This leads to the desired result since

$$\rho(\bar{x}, \bar{x}') \leq \frac{1}{nh(r,\kappa)}\left\|\exp_{\bar{x}'}^{-1}(x_n) - \exp_{\bar{x}'}^{-1}(x_n')\right\|_{\bar{x}'} \leq \frac{2r(2 - h(r,\kappa))}{nh(r,\kappa)},$$

using Lemma 1 in the Supplemental based on Jacobi field estimates [Karcher, 1977].

$\square$

In our next Theorem we provide a guarantee on the utility of our mechanism. We demonstrate that, in general, the magnitude of the privacy noise added is $O(dr/n\epsilon)$. Classic results on $\epsilon$-DP [e.g. Hardt and Talwar, 2010] in $\mathbb{R}^d$ typically do not calculate sensitivity based on a Euclidean ball, instead focusing on privatizing each coordinate separately, in which case the optimal privacy noise is $O(d^{3/2}r/n\epsilon)$. To reconcile the two, the classic rate can equivalently be thought of as calculating sensitivity using an $\ell_\infty$ ball. It is easy to verify that it requires an $\ell_2$ ball of radius $r\sqrt{d}$ to cover an $\ell_\infty$ ball of radius $r$, in which case the two rates agree, meaning that our mechanism is rate optimal.

**Theorem 3.** *Let the Assumptions of Theorem 2 hold. Let $\widetilde{x}$ denote a draw from the Laplace mechanism conditioned on being in $B_r(m_0)$. Assume that $n$ and $\epsilon$ are such that $\sigma \to 0$. Then $\widetilde{x}$ is $\epsilon$-DP and furthermore*

$$\mathrm{E}\,\rho(\widetilde{x}, \bar{x})^2 = O\left(\frac{d^2r^2}{n^2\epsilon^2}\right).$$

*Proof.* First, $\widetilde{x}$ conditioned on falling in $B_r(m_0)$ guarantees the existence of the Laplace distribution (over $B_r(m_0)$) since it is now clearly integrable. That $\widetilde{x}$ is DP follows from the same arguments as Theorem 1.

Turning to our utility guarantee, first notice that $r$ was chosen such that there exists a set $A_r \subset T_{m_o}$ such that the $\exp_{m_0} : A_r \to B_r(m_0)$ is a diffeomorphism. Identifying $T_{m_0}\mathcal{M}$ with $\mathbb{R}^d$, we also have that $A_r$ is a ball centered at $\mathbf{0}$ with radius $r$ (as measured using the inner product $\langle\cdot,\cdot\rangle_{m_0}$).

A change-of-variables, $\exp_{m_0}(\widetilde{v}) = \widetilde{x}$, implies that $\widetilde{v}$ has density equal to

$$f(v) = c_{f,\sigma}^{-1}e^{-|v|/\sigma}|J_v|$$

with support on $A_r$, where $|J_v|$ is the determinant of the Jacobian of $\exp_{m_0}$. This is not the K-norm distribution over $\mathbb{R}^d$ unless the Jacobian is constant in $v$. Since the set $A_r$ is compact, the determinant

of the Jacobian is bounded from above and from below (away from 0), we can find constants $c_1$ and $c_2$, independent of $\sigma$, satisfying

$$c_1 e^{-|v|/\sigma} \le e^{-|v|/\sigma} |J_v| \le c_2 e^{-|v|/\sigma}.$$

We can use this to bound the desired expected value as

$$\mathrm{E}\, \rho(\widetilde{x}, \bar{x})^2 = c_{f,\sigma}^{-1} \int_{A_r} |v|^2 e^{-|v|/\sigma} |J_v|\, dv \le \left[ c_1 \int_{A_r} e^{-|v|/\sigma}\, dv \right]^{-1} c_2 \int_{A_r} |v|^2 e^{-|v|/\sigma}\, dv.$$

Using a change of variables with $u = \sigma^{-1} v$ we have that

$$\mathrm{E}\, \rho(\widetilde{x}, \bar{x})^2 \le \frac{c_2 \sigma^2}{c_1} \left[ \int_{A_{r/\sigma}} e^{-|u|}\, du \right]^{-1} \int_{A_{r/\sigma}} |u|^2 e^{-|u|}\, du.$$

This, however, is simply $\sigma^2 2 c_2/c_1$ multiplied by the expected squared norm of a Euclidean Laplace that is conditioned on falling within $A_{r/\sigma}$. If we remove the condition that the Laplace falls within $A_{r/\sigma}$ the value necessarily increases, thus we have that

$$\mathrm{E}\, \rho(\widetilde{x}, \bar{x})^2 \le \frac{c_2 \sigma^2}{c_1} \left[ \int_{\mathbb{R}^d} e^{-|u|}\, du \right]^{-1} \int_{\mathbb{R}^d} |u|^2 e^{-|u|}\, du.$$

Using a change of variables in both integrals to spherical coordinates and noting $\sigma = O(r/n\epsilon)$ as in Theorems 1 and 2, we get that

$$\mathrm{E}\, \rho(\widetilde{x}, \bar{x})^2 \le \frac{c_2 \sigma^2}{c_1} \left[ \int_0^\infty y^{d-1} e^{-y}\, dy \right]^{-1} \int_0^\infty y^{d+1} e^{-y}\, dy = \frac{c_2 \sigma^2 d(d-1)}{c_1} = O\left( \frac{d^2 r^2}{n^2 \epsilon^2} \right).$$

$\square$

Our final Theorem focuses on the case of linear manifolds to highlight mathematically the benefit of constructing the privacy mechanism directly on the manifold as opposed to an ambient space and then projecting back onto the manifold. Practically, the variance of the mechanism is inflated by a factor of $D/d$ where $d$ and $D$ are the dimensions of the manifold and ambient space respectively with $d \le D$. Intuitively, if the ambient space is of a higher dimension than the manifold, then one has to expend additional privacy budget to privatise the additional dimensions, which our approach avoids. Since the mechanism concentrates around a single point as the sample size grows (and thus one can use a single tangent space to parameterise the problem), one should be able to extend this to more general manifolds, though for ease of exposition we focus on the linear case.

**Theorem 4.** *Let $\mathcal{M} \subset \mathbb{R}^D$ be a d-dimensional linear subspace of $\mathbb{R}^D$ equipped with the Euclidean metric. Assume the assumptions of Theorem 3 hold. Let $\widetilde{x}_D$ denote the private summary generated from the Laplace over $\mathbb{R}^D$ with scale $\sigma$, in the sense of Definition 3 (equivalently, this is the K-norm mechanism with the $\ell_2$ norm). Then*

$$\mathrm{E}\, \|\mathcal{P}_{\mathcal{M}} \widetilde{x}_D - \bar{x}\|^2 = O\left( \frac{dDr^2}{n^2 \epsilon^2} \right),$$

*where $\mathcal{P}_{\mathcal{M}}$ is the projection operator onto $\mathcal{M}$.*

*Proof.* Choose $\{v_1, \ldots, v_D\}$ as an orthonormal basis of $\mathbb{R}^D$ such that the matrix $\mathcal{P}_{\mathcal{M}} = VV^T$ is an orthogonal projector onto $\mathcal{M}$, where $V = [v_1, \ldots, v_d] \in \mathbb{R}^{D \times d}$ and $V^T V = \mathbb{I}_d$. Let $\langle \cdot, \cdot \rangle$ be the usual inner product on $\mathbb{R}^D$ with norm $\|\cdot\|$. Note that $\widetilde{x}_D = \bar{x} + \sigma R \mathbf{U}$, where $R$ distributed as Gamma$(D, 1)$, $\mathbf{U}$ is uniform on $\mathcal{S}^{D-1}$ (i.e. $\langle \mathbf{U}, \mathbf{U} \rangle = 1$) independent of $R$, and $\bar{x} \in \mathcal{M}$ (see Supplemental 1.4 for details). Then

$$\mathrm{E}\, \|\mathcal{P}_{\mathcal{M}} \widetilde{x}_D - \bar{x}\|^2 = \mathrm{E}\, \|\sigma R \mathcal{P}_{\mathcal{M}} \mathbf{U}\|^2 = \sigma^2\, \mathrm{E}[R^2] \sum_{i=1}^d \mathrm{E}[\langle \mathbf{U}, v_i \rangle^2].$$

Since $\mathbf{U}$ is uniform on the sphere, it follows that the vector $(\langle \mathbf{U}, v_1 \rangle^2, \ldots, \langle \mathbf{U}, v_D \rangle^2)$ follows a Dirichlet distribution with concentration parameters all equal to 1. Thus, for each $i$, $\langle \mathbf{U}, v_i \rangle^2$ is distributed as Beta$(1, D-1)$. This completes the proof as

$$\mathrm{E}\, \|\mathcal{P}_{\mathcal{M}} \widetilde{x}_D - \bar{x}\|^2 = d\sigma^2 (D + D^2) \frac{1}{D} = \sigma^2 d(D+1).$$

$\square$

# 5 Examples

In this section we numerically explore two examples that are common in statistics. In the first example, we consider the space of symmetric positive definite matrices (SPDM) equipped with the Rao-Fisher affine invariant metric, under which the space is a negatively curved manifold. In the second example we consider data lying on the sphere, which can be used to model discrete distributions or compositional data, as an example of a positively curved manifold. In both cases we demonstrate substantial gains in utility when privatizing the Fréchet mean using our proposed mechanism against using a more standard Euclidean approach that utilizes an ambient space. Other examples of Riemannian manifolds that commonly arise in statistics include the Steifel manifold for PCA projections, quotient spaces for modeling shapes, and hyperbolic spaces for modelling phylogenetic structures. Simulations are done in Matlab on a desktop computer with an Intel Xeon processor at 3.60GHz with 31.9 GB of RAM running Windows 10. Additional details on each example are also provided in the supplemental.

## 5.1 SPDM

Let $\mathbb{P}(k)$ denote the space of $k \times k$ symmetric positive definite matrices. In addition to being used for modeling covariance matrices, this space is widely used in engineering of brain-computer interfaces [Congedo et al., 2017], computer vision [Zheng et al., 2014], and radar signal processing [Arnaudon et al., 2013].

We consider $\mathbb{P}(k)$ equipped with the Riemannian metric $\langle v, u \rangle_p = \text{Tr}(p^{-1}up^{-1}v)$, known as the Rao-Fisher or Affine Invariant metric where $u, v \in T_p\mathbb{P}(k)$ are symmetric matrices. The metric makes $\mathbb{P}(k)$ into a manifold with negative sectional curvature [e.g., Helgason, 2001]. Under this metric $\exp_p(v) = p^{1/2}\text{Exp}\left(p^{-1/2}vp^{-1/2}\right)p^{1/2}$ and $\exp_q^{-1}(p) = q^{1/2}\text{Log}\left(q^{-1/2}pq^{-1/2}\right)q^{1/2}$, where Exp and Log are matrix exponential and logarithm respectively. The (squared) distance between $q, p \in \mathbb{P}(k)$ is given in closed form by $\rho^2(q, p) = \text{Tr}[\text{Log}(q^{-1/2}pq^{-1/2})^2]$; these expressions are widely available [e.g., Hajri et al., 2016, Said et al., 2017]. To calculate the Fréchet mean of a sample we use a gradient descent algorithm as proposed in Le [2001] (Supplemental material 1.1).

We generate samples $D = \{x_1, \cdots, x_{n-1}, x_n\}$ from $\mathbb{P}(k)$ using the Wishart distribution as discussed in the Supplemental material 1.2.1. In the first and second panels of Figure 1 we show simulation results which illustrate Theorems 2 and 3 and compare the utility of the Euclidean counterpart. In the first panel we illustrate the sensitivity by plotting $\rho(\bar{x}, \bar{x}')$, for neighboring databases, as blue dots as well as the theoretical bound The blue line is the average distance at each sample size.

We compute the Fréchet mean $\bar{x}$ and then privatize the mean using two approaches: (i) we generate the privatized mean $\tilde{x}$ by sampling using the Laplace distribution from Definition 3 defined directly on $\mathbb{P}(k)$ with footpoint $\bar{x}$ using the algorithm in Hajri et al. [2016]; (ii) we use the embedding of $\mathbb{P}(k)$ into the set of $k \times k$ symmetric matrices, isomorphic to $\mathbb{R}^{k(k+1)/2}$, to represent $\bar{x}$ as a vectorized matrix $\text{vech}(\bar{x})$ in $\mathbb{R}^3$ (i.e., $k = 2$) and obtain a privatized mean $\text{vech}(\tilde{x}_E)$ by adding to $\text{vech}(\bar{x})$ a vector drawn from the standard Euclidean Laplace. Then $\tilde{x}_E$ is obtained by reverting to the matrix representation, which is not guaranteed to stay in $\mathbb{P}(k)$ but is symmetric by construction.

In the second panel we plot the average, across repetitions, of the distances $\|\text{vech}(\bar{x}) - \text{vech}(\tilde{x})\|$ (blue) and $\|\text{vech}(\bar{x}) - \text{vech}(\tilde{x}_E)\|$ (red). Since the Euclidean summary need not belong to $\mathcal{M}$, using the Euclidean distance enables a common comparison between the two methods. The shaded regions around the lines correspond to $\pm 2\text{SE}$, where SE is the standard error of the average distances.

Examining panel 2 in Figure 1, we see that our approach has better utility even when calculated using the Euclidean distance. Here, the ambient space approach does not increase the dimension of the statistic since $\mathbb{P}(k)$ and the space of symmetric matrices are of the same dimension, thus the gain in utility appears to be primarily due to respecting the geometry of the problem. Furthermore, as expected for smaller sample sizes, approach (ii) can produce summaries that are not positive definite, with about 25% not being in $\mathbb{P}(k)$ at sample sizes 20-40.

## 5.2 Spheres

Let $\mathcal{S}_\kappa^d$ denote a $d$-dimensional sphere of radius $\kappa^{-1/2}$ parameterized such that the sectional curvature is constant $\kappa > 0$. Identifying the sphere as a subset of $\mathbb{R}^{d+1}$, the tangent space at $p$ is $T_p\mathcal{S}_\kappa^d =$

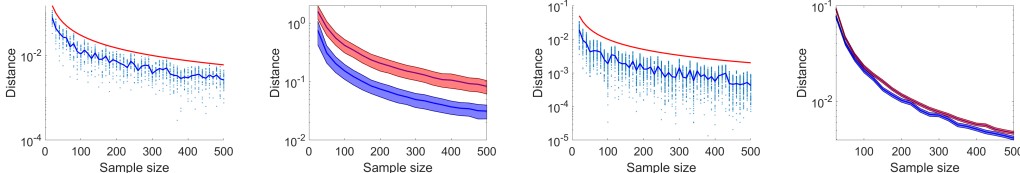

Figure 1: For the first and second panel are for $\mathbb{P}(2)$, while the third and fourth for $\mathcal{S}_1^2$. First and third panel: The blue points are the manifold distance between the Fréchet means $\bar{x}$ and $\bar{x}'$ of $D = \{x_1, \cdots, x_{n-1}, x_n\}$ and $D' = \{x_1, \cdots, x_{n-1}, x_n'\}$, respectively. The blue line is the average distance at each sample size and the red line is the theoretical bound on the sensitivity from Theorem 2. Second and fourth panel: At each sample size we generate several replicates (1000 for each $\mathbb{P}(2)$ and $\mathcal{S}_1^2$) and compute the Fréchet mean $\bar{x}$. We then separately privatize the Fréchet mean twice, first on the manifold which results in $\widetilde{x}$ and second embedding the mean onto Euclidean space, $\mathbb{R}^3$ in both cases, which results in $\widetilde{x}_E$. The blue bars represent the average of the Euclidean distances between $\bar{x}$ and $\widetilde{x}$ with the bounds $\pm$2SE; the red bars represent the average of the Euclidean distances between $\bar{x}$ and $\widetilde{x}_E$ with the bounds $\pm$2SE. For further detail see sections 5.1, 5.2, and the supplemental.

$\{v \in \mathbb{R}^{d+1} : \langle v, p \rangle = 0\}$. The exponential map, defined on all of $T_p\mathcal{S}_\kappa^d$, is given by $\exp_p(v) = \cos(\|v\|)p + \kappa^{-1/2}\sin(\|v\|)v/\|v\|$ and $\exp_p(\mathbf{0}) = p$. The inverse exponential map $\exp_p^{-1} : \mathcal{S}_\kappa^d \to T_p\mathcal{S}_\kappa^d$ is defined only within the ball of radius strictly smaller than $\pi/2$ around $p$ and is given by $\exp_p^{-1}(q) = \frac{\theta}{\sin(\theta)}(q - \cos(\theta)p)$ with $q \neq \{p, -p\}$. The corresponding distance function is $\rho(p, q) = \theta$ where $\theta = \cos^{-1}(\langle p, q \rangle)$ and $p, q \in \mathcal{S}_\kappa^d$.

In similar fashion to Section 5.1, in the third and fourth panels of Figure 1 we show simulation results which illustrate Theorems 2 and 3, and compare utility to its Euclidean counterpart. For our simulations we fix $d = 2$ and $\kappa = 1$. Consistent with Assumption 1 on support of the data, we choose a ball $B_r(m_0)$ of radius $r = \pi/8 < r^* = \pi/2$ and take $m_0$ as the north pole. We generate random samples as shown in the Supplemental material 1.3.1. The red line again corresponds to the theoretical bound from Theorem 2. The (unique) Fréchet mean $\bar{x}$ is computed using a gradient descent algorithm.

In the last panel we compare utility of the privatized means, again obtained using two approaches: (i) exactly as in approach (i) with SPDM; (ii) using the embedding of $\bar{x}$ into $\mathbb{R}^3$ (Cartesian coordinates) to represent $\bar{x}$ and obtaining a private $\widetilde{x}_E$ by adding to $\bar{x}$ a draw from the Euclidean Laplace. We display the average Euclidean distance between the mean and private mean $\pm$2SE, where SE is the standard error of the distances at each sample size; we use 1000 replicates at each sample size. The blue band is obtained using approach (i) and the red band using approach (ii). While the contrast between (i) and (ii) is not as stark as with SPDM, our approach still produces about 15% less noise, with an average of 16.8% reduction in the smaller sample sizes and 12% reduction in the larger sample sizes. Furthermore, unlike in the SPDM case, the Euclidean private summary is never on the manifold since $\mathcal{M}$ as a subset of $\mathbb{R}^{d+1}$ has measure zero.

## 6 Conclusions and Future Work

In this paper we have demonstrated how to achieve pure differential privacy over Riemannian manifolds by relying on the intrinsic structure of the manifold instead of the structure induced by a higher-dimensional ambient or embedding space. Practically, this ensures that the private summary preserves the same geometric properties as the non-private one. Theorem 3 shows that our mechanism matches the known optimal rates for linear spaces for the Fréchet mean, while Theorem 4 highlights the benefit of the intrinsic approach in contrast to projection-based ones using an ambient space.

The benefits of directly working on the manifold come at the expense of a more complicated mathematical and computational framework, challenges with which vary between different manifolds depending on availability of closed-form expressions for geometric quantities. Conversely, working in a linear ambient space is usually computationally simpler, although other issues abound: projecting onto the manifold may not be possible (e.g. SPDM under negative curvature), or the projection may lead to poor utility in high-curvature places on the manifold.

As is well appreciated in the mathematics/statistics literature, there are challenges that are unique to positively curved spaces, which we also encounter here. The central issue is that the squared distance function need not be convex over large enough areas, and strong restrictions on the spread of the data are required to ensure that underlying summaries are unique and well defined. This phenomenon manifests in the form of a correction term $h$ in our results, which impacts sensitivity of the statistics. Numerical illustrations in Section 5 show that this is not just a technical oddity or gap in our proofs since our empirical sensitivity can be seen to be quite close to our theoretical bound.

As this is the first paper we are aware of in DP over general manifolds, there are clearly many research opportunities. A deeper exploration over positively curved spaces would be useful given how common they are in practice (e.g., landmark shape spaces). A class of spaces unexplored in this paper, but well worth investigating, are Hadamard or (complete) CAT(0) spaces. They are non-positively curved metric spaces that need not be manifolds, on which geodesics and Fréchet means can be computed, and represent a natural geometric setting for graph and tree-structured data. One could also extend any number of privacy tools to manifolds including the Gaussian mechanism, exponential mechanism, objective perturbation, K-norm gradient mechanism, approximate DP, concentrated DP, and many others.

## Acknowledgments and Disclosure of Funding

This work was funded in part by NSF SES-1853209, the Simons Institute at Berkeley and their 2019 program on Data Privacy to MR; and, NSF DMS-2015374, NIH R37-CA214955 and EPSRC EP/V048104/1 to KB. We thank Huiling Le for helpful discussions.

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
