# Supplemental to *Differential Privacy Over Riemannian Manifolds*

## 1 Simulation details

Simulations are done in Matlab on a desktop computer with an Intel Xeon processor at 3.60GHz with 31.9 GB of RAM running Windows 10.

### 1.1 Computing the Fréchet mean

We use a gradient descent algorithm to compute the Fréchet mean of a sample $D = \{x_1, x_2, \ldots, x_n\}$. We initialize the mean $\hat{\mu}_0$ at any data point, take a small step in the average direction of the gradient of energy functional $F_2 : \mathcal{M} \to \mathbb{R}$, and iterate. If $\hat{\mu}_{k-1}$ is the mean in the $(k-1)$th iterate, in normal coordinates, the gradient is given by $v_k = \frac{1}{n} \sum_{i=1}^{n} \exp_{\hat{\mu}_{k-1}}^{-1}(x_i)$. Then, the estimate of the Fréchet mean at iterate $k$ is $\hat{\mu}_k = \exp_{\hat{\mu}_{k-1}}(t_k v_k)$ where $t_k \in (0, 1]$ is the step size. The algorithm is assumed to have converged once the change in the mean across subsequent steps is no longer significant, measured using the intrinsic distance $\rho$ on $\mathcal{M}$; that is, the algorithm terminates if $\rho(\mu_k, \mu_{k-1}) < \lambda$ for some pre-specified $\lambda > 0$. We choose the step size $t_k = 0.5$ and $\lambda = 10^{-5}$. In addition, one could set a maximum number of iterations for situations when the mean oscillates between local optima, and we set this at 500 but note that in our settings the algorithm typically converges in fewer than 200 iterations.

### 1.2 SPDM simulations

#### 1.2.1 Generating random samples within $B_r(m_0)$

Since $\mathbb{P}(k)$ can be identified with the set of $k \times k$ covariance matrices, we use the Wishart distribution $W(V, df)$ to generate samples from $\mathbb{P}(k)$, parameterized by a scale matrix $V$ and degrees of freedom $df > 0$ such that if $X \sim W(V, df)$ then $EX = df\, V$. We set $V = \frac{1}{k} I_k$, where $I_k$ is the identity matrix and $df = k$.

Recall that under the Rao-Fisher affine-invariant metric $\mathbb{P}(k)$ is negatively curved and thus the radius $r$ of ball $B_r(m_0)$ in Assumption 1 within which data $D$ is assumed to lie in is unconstrained. This results in samples from $W(\frac{1}{k} I_k, k)$ being centred at $I_k$, which can be viewed as a suitable value for $m_0$. It is possible, however, that certain samples from $W(\frac{1}{k} I_k, k)$ are at a distance (in terms of $\rho$) greater than $r$ from $m_0 = I_k$ since the affine-invariant metric

is not used in the definition of the Wishart, but this can always be adjusted by selecting a suitable radius $r$. A sample of size $n$ is thus generated by: (i) sampling $X \sim W(\frac{1}{k}I_k, k)$; (ii) retaining $X$ if $\rho(X, I_k) < r$ or (re-)sampling $X$ until distance is smaller than $r$. For our simulations, we set $r = 1.5$ and $df = 2$.

### 1.2.2 Radius in ambient space of symmetric matrices

Let $Sym_k$ be the set of $k \times k$ symmetric matrices within which $\mathbb{P}(k)$ resides. In order to compare sensitivity of the proposed method using geometry of $\mathbb{P}(k)$ to one which considers only the ambient space $Sym_k$, the radius $r_E$ of a ball in $Sym_k$, with respect to the distance induced by the Frobenius norm $\|\cdot\|_2$, that in a certain sense 'corresponds' to $r$ on $\mathbb{P}(k)$ needs to be ascertained.

This amounts to determining how the distance $\rho(x, y) = \|\mathrm{Log}(x^{-1/2}yx^{-1/2})\|_2$ under the affine-invariant metric on $\mathbb{P}(k)$ compares to $\|x - y\|_2$ when $x, y \in \mathbb{P}(k)$. Since $m_0 = I_k$, we can choose $y = I_k$ and compare $\|\mathrm{Log}(x)\|_2$ with $\|x - I_k\|_2$.

In particular, we seek to find the smallest Euclidean ball in $Sym_k$ that contains the geodesic ball $B_r(I_k)$ in $\mathbb{P}(k)$, and we accordingly define the radius of the Euclidean ball $r_E$ to be

$$r_E = \sup_{x \in \mathbb{P}(k): \|\mathrm{Log}(x)\| \leq r} \|x - I\|.$$

Expressing $x$ in its diagonal basis following a suitable change of coordinates leaves $\|\mathrm{Log}(x)\|$ and $\|x - I\|$ unchanged, and hence

$$r_E^2 = \sup_{\lambda: \sum_i \log(\lambda_i)^2 \leq r^2} \sum_i (\lambda_i - 1)^2,$$

where $\lambda_i > 0, i = 1, \ldots, k$ are the eigenvalues of $x$.

**Proposition 1.** $r_E = e^r - 1$.

*Proof.* Let $u_i = \log \lambda_i, i = 1, \ldots, k$. Consider the value of the objective function with the vector

$$(u_1, u_2, u_3, \ldots, u_k) = \left( \sqrt{u_1^2 + u_2^2}, 0, u_3, \ldots, u_k \right).$$

The reason behind assuming such a structure for the vector of eigenvalues is that if we can show that the value of objective function increases by replacing $(u_1, u_2)$ by $(\sqrt{u_1^2 + u_2^2}, 0)$, then by symmetry the objective function will be maximized by placing all of the weight in the first coordinate and setting all remaining coordinates to $\lambda_i = 1$. Consider a Taylor expansion of each of the terms (assume wlog that $u_i \geq 0$):

$$(e^{u_i} - 1)^2 = \left( \sum_{n=1}^{\infty} \frac{u_i^n}{n!} \right)^2 = \sum_{n,m} \frac{u_i^{n+m}}{n!m!} \tag{1}$$

versus

$$(e^{\sqrt{u_1^2+u_2^2}} - 1)^2 = \left( \sum_{n=1}^{\infty} \frac{(u_1^2+u_2^2)^{n/2}}{n!} \right)^2 = \sum_{n,m} \frac{(u_1^2+u_2^2)^{(n+m)/2}}{n!m!}. \qquad (2)$$

We wish to establish that

$$\sum_{n,m} \frac{(u_1^2+u_2^2)^{(n+m)/2}}{n!m!} \geq \sum_{n,m} \frac{u_1^{n+m} + u_2^{n+m}}{n!m!} \; .$$

This will hold if

$$(u_1^2+u_2^2)^{(n+m)/2} \geq u_1^{n+m} + u_2^{n+m},$$

or equivalently

$$u_1^2 + u_2^2 \geq \left( (u_1^2)^{(n+m)/2} + (u_2^2)^{(n+m)/2} \right)^{2/(n+m)}.$$

However, since $n+m \geq 2$, this follows immediately from the triangle inequality for $\ell_p$ spaces. □

## 1.3   Sphere simulations involving spheres

### 1.3.1   Generating samples within $B_r(m_0)$

We use polar coordinates to sample from a $d = 2$-dimensional sphere $\mathcal{S}_1^2$ of radius one (thus $\kappa = 1$). Let $(\theta, \phi)$ be the pair of polar and azimuthal angles, respectively, where $\theta \in [0, \pi]$ and $\phi \in [0, 2\pi)$. We uniformly sample on $\theta \in [0, r]$ and $\phi \in [0, 2\pi)$ with $r = \pi/8$. This results in data concentrated about the north pole ($m_0$), with increasing concentration towards the north pole.

### 1.3.2   Radius in ambient space

Suppose we have a dataset $D$ on the unit sphere, $\mathcal{S}_1^d$, centered at $m_0$ with (manifold) radius $r$. As with SPDM, we need to determine a suitable ball of radius $r_E$ in Euclidean space that contains the geodesic ball of radius $r$ on $\mathcal{S}_1^d$. To determine $r_E$, we simply need to convert $r$, which corresponds to arc length, to the chord length (from $m_0$ to the boundary of the ball). That is, since the sphere radius equals 1, $r_E = 2\sin(\frac{r}{2})$. With $r = \pi/8$ we obtain $r_E = 2\sin(\pi/16)$.

## 1.4   Sampling from the Euclidean Laplace

We discuss how to sample from K-norm mechanism with the $\ell_2$ norm $|\cdot|_2$ (i.e. the Euclidean Laplace) on $\mathbb{R}^d$. First, wlog we can take $\bar{x} = 0$ and $\sigma = 1$ as we can clearly generate from a standardized distribution and then translate/scale the result. So the goal is to sample from

$$f(y) \propto e^{-|y|_2}.$$

Evidently, $f$ is a member of the elliptical family of distributions. We thus sample a vector $y = (y_1, \ldots, y_d)$ from $f$ by sampling a direction uniformly on the $(d-1)$-dimensional unit sphere $\mathcal{S}_1^{d-1}$ and then, independently, sampling a radius $r > 0$ from an appropriate distribution. To determine this distribution, let

$$y_1 = r\cos(\theta_1)$$
$$y_2 = r\sin(\theta_1)\cos(\theta_2),$$
$$\vdots$$
$$y_{d-1} = \sin(\theta_1)\sin(\theta_2)\ldots\cos(\theta_{d-1})$$
$$y_d = \sin(\theta_1)\ldots\sin(\theta_{d-1}).$$

Then the density $f$ assumes the form

$$f(r, \theta_1, \ldots, \theta_d) = e^{-r}r^{d-1}\sin^{d-2}(\theta_1)\ldots\sin(\theta_{d-1}).$$

Since $f$ factors into a function of $r$ and a function of the angles, the distribution of $r$ is proportional to $r^{d-1}e^{-r}$, which is just the Gamma distribution $\Gamma(d, 1)$ with parameters $\alpha = d$ and $\beta = 1$. Thus, to sample a value from $f$:

1. sample a direction $U$ uniformly from $\mathcal{S}^{d-1}$;

2. sample a radius $R$ from $\Gamma(d, 1)$ distribution;

3. set $Y = \bar{x} + R\sigma U$.

Then $Y$ will be a draw from the $d$-dimensional Euclidean Laplace with scale $\sigma$ and center $\bar{x}$. In order to sample $U$ from $\mathcal{S}_1^{d-1}$, we use the well-known fact that if $X \sim N_d(\mathbf{0}_d, I_d)$ then $U := X/|X|_2$ follows a uniform distribution on $S^{d-1}$.

## 1.5 Sampling from the Laplace on the sphere $\mathcal{S}_1^d$

To sample from the Laplace on $\mathcal{S}_1^d$ we generate a Markov chain by using a Metropolis-Hastings random walk. At each step $n$ we generate a proposal $x'$ by first randomly drawing a vector $v$ in the current tangent space, $T_{x_n}\mathcal{S}_1^d$, then move on the sphere using the exponential map and said vector, $f(x'|x_n) = \exp_{x_n} v$. To draw $v$ we uniformly sample on a ball centered at the current step $x_n$ with radius $\sigma$ by drawing a vector from $N_3(0_3, I_3)$, scaling the resulting vector to have length $\sigma$, and projecting the vector onto the tangent space of $x_n$. This projection ensures that $\|v\| \leq \sigma$, which we take to be much smaller than the injectivity radius. Further, $v$ is not uniform in $T_{x_n}\mathcal{S}_1^d$, but since we use a Metropolis-Hastings algorithm, we only require symmetry is satisfied. One can sample vectors on the required tangent space in several manners, the proposed method is chosen for computational ease.

We aim to accept/reject draws from $f$ to produce a Markov chain with stationary density $C_{\eta,\sigma}^{-1}\exp(-\rho(\eta, x)/\sigma)$, where $\rho$ is the arc distance on the sphere. We follow a standard Metropolis-Hastings schematic.

1. Initialize $x_0 = \eta$.

2. In the $n$th iteration, draw a vector in $T_{x_n}\mathcal{S}_1^d$, the tangent space of $x_n$, as described earlier denoted as $v$.

3. Generate a candidate $x'$ by letting $x' = \exp_{x_n} v$.

4. Accept $x'$ and set $x_{n+1} = x'$ with probability $\exp(-\rho(\eta, x')/\sigma)/\exp(-\rho(\eta, x_n)/\sigma)$. Otherwise, reject $x'$ and generate another candidate by returning to previous step.

5. Return to step 2 until one has generated a sufficiently long chain.

The final sample is chosen based on a burn-in period of 10 000 steps and jump width of 100 to avoid correlated adjacent steps in the chain.

## 2 Bounding Distances on the Tangent Space

To complete Theorem 2 we need a bound on the distance

$$\| \exp_m^{-1}(x) - \exp_m^{-1}(y)\|_m,$$

which holds uniformly across all $m, x, y \in B_r(m_0)$; in particular, we seek a Lipschitz bound that holds uniformly over $B_r(m_0)$.

**Lemma 1.** *Under the assumptions of Theorem 2, for $x, y, m \in B_r(m_0)$ we have*

$$\| \exp_m^{-1}(x) - \exp_m^{-1}(y)\|_m \leq 2r(2 - h(r, \kappa)).$$

*Proof.* We first establish that the inverse exponential map at a fixed $m \in B_r(m_0)$ is Lipschitz. The map $\exp_m : T_x M \to \mathcal{M}$ when restricted to a ball $B$ of radius $r$ around the origin is a diffeomorphism since inj $m < r$. The inverse $\exp_m^{-1}$ is differentiable on $\exp_m(B)$. Let $N_m = \exp_m(B) \cap B_r(m_0)$ and consider a minimizing geodesic $\gamma : [0, 1] \to \mathcal{M}$ starting at $m$ that lies entirely in $N_m$.

The derivative $\mathrm{D}\exp_m^{-1}$ is the value of $\dot{J}(1) = \frac{\mathrm{D}}{ds}\frac{d}{dt}c_m(s, t)\Big|_{s=1}$, where $J(s) = \frac{d}{dt}c_m(s, t)$ is the Jacobi field along $s \mapsto c_m(s, t) = \exp_m\{s\exp_m^{-1}(\gamma(t))\}$, with $J(0) = 0$ and $J(1) = \dot{\gamma}(t)$. When restricted to the (compact) closure of $B$, the map $v \mapsto \|\mathrm{D}\exp_m^{-1} v\|_m$ is bounded above by $C(m) < \infty$.

Let $\Gamma := \exp_m^{-1}(\gamma)$. Comparing lengths of $\gamma$ and $\Gamma$ we obtain

$$L(\Gamma) \leq C(m) \int_0^1 \|\gamma'(t)\|_{\gamma(t)}\mathrm{d}t = C(m)L(\gamma),$$

since $\|\cdot\|_m$ is continuous. Since distances are obtained by minimising lengths of paths of curves, $\exp_m^{-1}$ is Lipshcitz with constant $C(m)$ on $N_m$. However, under

the assumptions of Theorem 2, from Jacobi field estimates A5.4, when used in conjunction with Corollary 1.6, in [1], we get

$$C(m) \leq \sup_{v \in T_m M, \|v\|_m = 1} \|\mathrm{D}\exp_m^{-1} v\|_m \leq \begin{cases} 2 - h(r, \kappa) & \text{if } \kappa > 0 \\ 1 & \text{if } \kappa \leq 0, \end{cases}$$

where $h(r, \kappa)$ is as defined in Theorem 2. As a consequence,

$$\|\exp_m^{-1}(x) - \exp_m^{-1}(y)\|_m \leq [2 - h(r, \kappa)]d(x, y) \leq 2r[2 - h(r, \kappa)],$$

as desired. □

## 2.1 An Empirical Bound on the Sensitivity for $\mathcal{S}_1^d$

The sensitivity is bound as $\rho(\bar{x}, \bar{x}') \leq \frac{2r(2 - h(r, \kappa))}{nh(r, \kappa)}$ where $h(r, \kappa)$ is a function of the radius of the ball $B_r(m_0)$ and $\kappa$ the sectional curvature of the manifold. The correction factor in the numerator, which is only present for positively curved manifolds, is not very tight for large radius ball. From the theorem we see that this correction factor comes from $\|\exp_m^{-1}(x) - \exp_m^{-1}(y)\| \leq 2r[2 - h(r, \kappa)]$, so we consider this norm in the case of the unit sphere.

Given a ball $B_r(m_0)$ and any three points $x_1, x_2, x_3 \in B_r(m_0)$, we wish to find an empirical bound on $\|\exp_{x_1}^{-1}(x_2) - \exp_{x_1}^{-1}(x_3)\|$. To do this we create a uniformly spaced grid on the boundary of $B_r(m_0)$ to produce a set $\{x_i\}$. We then fix an arbitrary point, say $x_1$, and compute $\max_{i,j} \|\exp_{x_1}^{-1}(x_i) - \exp_{x_1}^{-1}(x_j)\|$. Because of the symmetry of the ball on the sphere, one can fix the footpoint and search over all other points. In Figure 1 we display the radius of $B_r(m_0)$ as the x-axis, $2r$ in blue, $2r[2 - h(r, \kappa)]$ in red, and $\max_{i,j} \|\exp_x^{-1}(x_i) - \exp_x^{-1}(x_j)\|$ in yellow. We see that $2r < \max_{i,j} \|\exp_{x_1}^{-1}(x_i) - \exp_{x_1}^{-1}(x_j)\| < 2r(2 - h(r, \kappa))$ however the inflation due to the the curvature is not as large as our bound. Rather than use the theoretical bound in the simulations of the sphere, we use the empirical bound.

---

[1]H. Karcher. *Riemannian center of mass and mollifier smoothing*, Communications in Pure and Applied Mathematics (1977), 509-541.

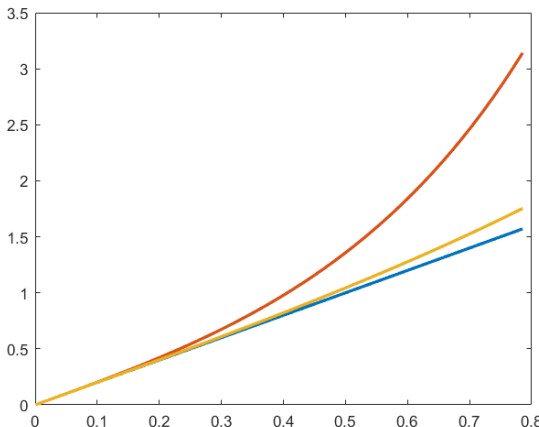

Figure 1: The x-axis represents the radius of $B_r(m_0)$. The blue line is $2r$, the red line is $2r(2 - h(r, \kappa))$, and the yellow line is $\max_{i,j} \| \exp_x^{-1}(x_i) - \exp_x^{-1}(x_j) \|$.