# OpenReview forum: "Differential Privacy Over Riemannian Manifolds"
_NeurIPS.cc/2021/Conference — NeurIPS 2021 Poster_

### Official Review · Reviewer_f85U · 2021-07-06

**Rating:** 9
**Confidence:** 3

**Summary:**

This paper studies differential privacy for the case of data and/or summary which consists of points in a Riemannian manifold.
The Laplace mechanism for DP is adapted to this case, basic results on its effectiveness are proven, and two examples are worked out.

**Limitations And Societal Impact:**

Differential privacy is a topic which involves public purpose considerations.
But the present work is abstract and does not directly touch on them.

**Main Review:**

This paper studies differential privacy for the case of data and/or summary which consists of points in a Riemannian manifold.
After pointing out that the standard definition generalizes, several simple and elegant results are proven:
1. The Laplace mechanism for realizing DP generalizes straightforwardly.
2. The sensitivity of the Frechet mean is bounded, so it can be used to produce a DP summary.
3. The intrinsic definition is shown to work better than the definition by embedding the manifold.

Then, two simple examples are studied numerically, one with negative curvature (symmetric positive definite matrices) and one with positive curvature (the sphere), illustrating the bounds and comparisons.

This is a very nice work and can also serve as an introduction to DP for non-experts (such as myself).
I cannot judge the authors' claim that this is the first work on DP for manifolds, but granting it,
this seems quite suitable for NeurIPS.

**Time Spent Reviewing:**

1

---

> ### Author Response · Authors · 2021-08-10
> **Response to f85U**
>
> Thank you, we appreciate your time and the comments.

---

### Official Review · Reviewer_q1gp · 2021-07-12

**Rating:** 7
**Confidence:** 3

**Summary:**

There are two main contributions in the paper:
- First, the language of differential privacy is adapted to the framework of Riemannian manifolds. In particular, an adaption of the Laplace mechanism on manifolds is proposed.
- Second, the Laplace mechanism is applied to produce differentially private Fréchet means on a manifold. The key result to do so is to provide a stability result for Fréchet means. The curvature of the manifold is shown to play a central role in this stability property.

**Ethical Concerns:**

No ethical concerns

**Limitations And Societal Impact:**

Yes

**Main Review:**

The different ideas developed in this paper all feel very natural, while it seems they were missing from the literature so far. I particularly like the stability theorem (Theorem 2) that highlights the impact the geometry of the manifold (namely its curvature) has on the sensitivity of the Fréchet mean. What is gained by working directly on the manifold (instead of working in the ambient space) is also showcased by Theorems 3 and 4, both results being convincing. Even if the two numerical experiments are compelling, a supplementary experiment involving a non-synthetic dataset lying on a manifold would be welcome.

Minor comments:

l.93: is the supremum of... I do not understand this definition.

l.133: define what the Laplace mechanism is

l.145: to be A global minimizer

l.151: I think it should be $x_i = 2\pi i/(n-1)$

l.152: minimizing the ENERGY FUNCTIONAL $F_2$

l.160: The data...

l.162: " it suffices for radius "

l.167: I could not find this assertion in [Le, 2001]

l.173: depends only on $r$ and ON...

l.181: for THE datasets

l.183: "under Assumption 1 the log map or inverse exponential map $y\in M\mapsto \exp^{-1}(y)\in T_xM$ is well-defined for every $x\in M$" -> the log map is defined on a geodesic ball.

l.189: $G_2$ instead of $G$

l.190: 0<t_0<b

l.193: what is the definition of z?

l.201: I think it should be $1$ for $\kappa=0$.

l.202: Replace with "If $\kappa>0$, the function $s\mapsto a(s,\kappa)$ is decreasing and upper bounded by $1$" (or similar formulation)

l.217: specify what is $\sigma$ in the Laplace mechanism

l.250: we want that the expected distance is larger than the quantity (and not a big O)

l.256: I could not find the details in the supplementary material.

l.264: THE Rao-Fisher metric

**Time Spent Reviewing:**

4

---

> ### Author Response · Authors · 2021-08-10
> **Response to q1gp**
>
> We appreciate the comments and your time.  We agree that a real data example would be compelling.  Instead of hastily putting together an application, we will add several references to the introduction that highlight some of the applications of manifolds in statistics.  We also appreciate all of the minor comments, which we will fix.

---

### Official Review · Reviewer_62w6 · 2021-07-16

**Rating:** 6
**Confidence:** 4

**Summary:**

This paper is concerned with extending the Laplace mechanism (and its guarantees) beyond Euclidean spaces, in order to achieve pure differential privacy over manifolds without resorting to first embed the space into an Euclidean space. This seems like an interesting idea, which can lead to some savings in the parameters (by constant factors, see below) at the price of some computational issues and (as far as I can tell) a more restrictive assumption on the data (Assumption 1).

The authors specifically consider the question of bounding the sensitivity of the Frechet mean (Theorem 2), in order to compute privately (pure privacy) the Frechet mean of a dataset (Theorem 3) matching the corresponding guarantee for Euclidean spaces.

**Limitations And Societal Impact:**

I believe the authors did adequately address them. For a work of that theoretical nature, the societal impacts are very far down the line, and assessing them amounts (with a little hyperbole) to asking what Peano axioms might imply for the trolley problem.

**Main Review:**

This generalization from Euclidean spaces to manifolds looks worth investigating to me, and the fact that in under the assumptions made the privacy overhead matches the special case of Euclidean spaces is appealing. However, I have some reservations about the paper, given that (besides the technical notions and proofs required to handle the more general setting) there isn't a lot of conceptually surprising content, as far as I can tell:
- the generalizations of the Laplace mechanism, and sensitivity, are the direct and natural analogues of the Euclidean case
- the savings compared to the naive embedding approach do not seem incredible (constant factors), at least for the example discussed (see below)
- the computational aspects are evoked, but would gain to be explored in more detail
- the future directions mentioned in ll 357-359 should, I believe, have been at least partly explored in this very paper: especially exponential and Gaussian mechanisms. Would the techniques required be that different?

As such, I don't believe the paper, as it stands, reaches the bar for NeurIPS, though it may be well suited for a less competitive venue.

- How computationally easy is Assumption 1 to check (find the optimal r) given a manifold and a dataset?
- If "minimax rates in DP typically scale polynomially in the dimension" and "the Whitney embedding theorem states that, in the worst case, to embed a manifold in Euclidean space requires a space that is twice the dimension of the manifol", then your approach here only saves constant factors, which is not a very strong motivation, especially given that you have to impose additional restricitions (Assumption 1) and may lose constant factors elsewhere (there are other motivations the authors put forward, but this saving is one of the ones put front and center, in p.1).

Maybe look for a setting/estimation problem where the rate scales superpolynomially in the dimension? Then you would get a much clearer advantage.

Minor points:

Theorem 1: "foot-point" -> footpoint
- Put the definition of h in Theorem 2 or closer to the statement theorem, not a page later. Also, maybe plot it to have a sense of its behavior? For instance, recalling in the theorem, after that, that sr sqrt(kappa) is < pi/2 by Assumption 1 might be helpful, since that's the thing that ensures h > 0

====
Updated my score in view of the authors' response and reviewers' discussions.

**Time Spent Reviewing:**

2

---

> ### Author Response · Authors · 2021-08-10
> **Response 62w6**
>
> Thank you for your time and the comments.  We agree that our extension to manifolds seems natural, however, we view this as a positive and don't view it as detracting from the novelty of the work; we still aren't aware of other DP works that consider general manifolds. {Indeed, since locally every manifold is Euclidean, it is reasonable to expect natural counterparts of certain quantities in DP that depend only on local properties of chosen statistics. We view a major contribution of this work as not only validating such an expectation but also in demonstrating that even certain global properties such as sensitivity can be handled satisfactorily under some additional conditions on curvature of the manifold, which is an inherently local property}.
>
> We believe that the reviewer isn't appreciating the savings demonstrated by our approach. Minimax rates in DP are often expressed as functions of the privacy budget, the dimension, and the sample size; so calling the savings a `constant factor' is not accurate.  More practically, getting to save a fraction of the privacy budget by not sanitizing extra ambient dimensions seems worthwhile. Secondly, asymptotic rates ignore the fact that our methods can preserve important geometric properties of the statistical summaries that might not even be possible (or at least very cumbersome) using classic methods, e.g., making a symmetric matrix positive definite and not just nonnegative definite.
>
> We agree that it would be interesting to include more details on the computational side.  However, this can depend greatly on the manifold.  For example, for our positive definite matrices, we used a nearly closed form sampler to draw from the Laplace distribution, while for the sphere we were unaware of any closed form sampler and had to rely on rejection sampling. {This is related to the fact that the set of positive definite matrices can be made negatively curved under a certain metric, while the same is not possible for a two-dimensional sphere. Developing sampling algorithms on manifolds is an active area of research, and challenges involved are intimately tied to (1) whether it is viewed as an embedded manifold, (2) choice of metric and (3) corresponding sectional curvatures. Consequently, there are very few works that address the sampling problem in any level of generality (See, for e.g., the paper `Sampling from a manifold' by Diaconis, Holmes and Shahshahani, Institute of Mathematical Statistics Collections, 2013: 102-125 (2013)).}
>
> The extensions mentioned would seem to require quite a bit of additional work, especially the Gaussian mechanism (which is not pure dp, only approximate, i.e. $(\epsilon,\delta)$ DP) where controlling the $\delta$ could get delicate. {Regarding Assumption 1, we first note that the radius is unconstrained if the manifold is negatively curved (hyperbolic spaces). For positively curved manifolds,  Assumption 1 is not especially difficult to verify if one can compute the exponential map.  This kind of assumption is necessary for most DP methods since the influence of a single data point has to be bounded in some way to control the sensitivity.  The only additional challenge is computing the injectivity radius and an upper bound on sectional curvatures, which again varies by manifold. For manifolds of constant curvature the injectivity radius can be computed. These include some popularly used manifolds in statistics such as spheres, hyperbolic spaces and shape space of two-dimensional landmarks used in shape analysis. For general manifolds, bounds on the injectivity radius can be obtained if volumes of specifically defined balls can be approximated (see, for e.g., Section 4 of the paper `Finite propagation speed, kernel estimates for functions of the Laplace operator, and the geometry of complete Riemannian manifolds' by Cheeger, Gromov and Taylor, J. Differential Geom. 17(1): 15-53 (1982)). We will add a few lines discussing these matters following Assumption 1 in the revision.}
>
> Finding an example where one gets more severe scaling in the dimension (e.g. nonparametric density estimation), would be interesting, but would have to occur in a follow up paper as it would be a substantial undertaking.  We will add some discussion and references highlighting how the scaling in the dimension can be much worse in certain statistical and machine learning problems (e.g. Stone (1980), Korostelev and Tsybakov (1993)).
>
> We agree with the minor comments and will make the adjustments.

---

> > ### Comment · Reviewer_62w6 · 2021-08-20
> > **Thank you for your response...**
> >
> > ... after reading it, and discussions with the other reviewers, I realize that my assessment re: constant savings was quite harsh (I also didn't pay enough attention to the experiments, focusing on the asymptotic, O()-type statements, which effectively hide/nullify any constant savings).
> >
> > I would however recommend detailing a bit more that point, especially in the intro which mentions the Whitney embedding theorem off the bat.
> >
> > I have still some reservations, but much milder than originally and have updated my score accordingly.

---

> > > ### Author Response · Authors · 2021-08-23
> > > **Response 62w6 - 2**
> > >
> > > We greatly appreciate it.  We agree that the point about the nature of the asymptotic savings could be made more clear and will adjust the paper accordingly.

---

### Official Review · Reviewer_MJ89 · 2021-07-17

**Rating:** 9
**Confidence:** 3

**Summary:**

In this paper, the authors consider the problem of releasing a differentially private statistical summary that resides on a Remannian manifold. They present an extension of the Laplace or K-norm mechanism that utilizes intrinsic distances and volumes on the manifold.

**Limitations And Societal Impact:**

Yes. I think this paper is good.

**Main Review:**

They consider in detail the specific case where the summary is the Fr\'echet mean of data residing on a manifold and demonstrate that the mechanism is rate optimal and depends only on the dimension of the manifold, not on the dimension of any ambient space, while also showing how ignoring the manifold structure can decrease the utility of the sanitized summary.

**Time Spent Reviewing:**

10

---

> ### Author Response · Authors · 2021-08-06
> **Response to MJ89**
>
> Thank you, we appreciate your time and comments.

---

### Official Review · Reviewer_aauF · 2021-07-26

**Rating:** 5
**Confidence:** 3

**Summary:**

This paper studies the problem of releases a differentially private statistic when the data and the statistic lie on a Riemannian manifold. They give a formulation of the Laplacian mechanism in this setting. They then consider the specific example of.a Frechet mean to display the benefits of using the manifold structure rather than adding noise in the ambient (higher dimensional) space, then projecting back to the manifold.

The authors show through specific examples and experiments the benefit of using the manifold structure rather than adding noise in the ambient space. As a first paper on the topic this seems like a nice proof of concept. I would have liked it to go deeper in discussing the complexity of actually sampling from the Laplace mechanism on a manifold. In particular, assumption 1 felt like it simplified the problem considerably.

**Ethical Concerns:**

I do not have any ethical concerns.

**Limitations And Societal Impact:**

I do not see any negative societal impact.

**Main Review:**

Developing a toolbox for DP on manifolds is an interesting problem. There are many problems where some inherent structure is known (the authors give the examples of covariance matrices and discrete distributions) and previous papers in the DP literature have expended considerable effort to design algorithms with error scaling with the dimension of the problem, not the ambient dimension. The paper is largely well written (some additional comments on presentation can be found below). The authors do a good job of motivating DP on manifolds.

Although there are papers in the DP literature that utilise the underlying manifold structure of a problem, I am not aware of any papers that attempt to give a general treatment or general purpose algorithms.

One of the main contributions is the definition of the Laplace distribution on a manifold. I thought this section needed more discussion. For example is the mean of this distribution the foot-print f(D)? What are the tails? What does it look like on different manifolds?

My biggest concern with the paper comes in assumption 1 in the Frechet mean section, which assumes all the data is contained in a set that is diffeomorphic to a compact subset of Euclidean space. The authors discuss why the radius r* is defined how it is, but I was still unsure about why simpler solutions are ineffective in this setting? In particular, why can’t one push the entire computation into the tangent plane at m_0 then just use the regular Laplace mechanism? This seems particularly pertinent given that the authors rely later on the fact that this assumption implies that the determinant of the Jacobian is a constant.

The baseline comparisons in the experiments seem a little strange since adding noise in the ambient space is not a state of the art DP algorithm for these problems.


==== Minor Comments

While I understand that an introduction to manifolds is outside the scope of this paper, I think the readership could be broadened by adding a simple graphic explaining the tangent plane, exponential map, geodesics and Riemannian volume intuitively.

It was a little unclear to me when reading assumption 1 whether (there exists r and m_0 such that for all databases D, D \subset B_r(m_0)) or (for all databases D, there exists r and m_0…)

Something seems amiss about the second sentence of the proof of Theorem 2. Assumption 1 doesn’t seem to imply anything about the manifold itself.


**Time Spent Reviewing:**

4

---

> ### Author Response · Authors · 2021-08-10
> **Response to aauF**
>
> The mean of the Laplace distribution need not be the footpoint.  We agree that it would be interesting to explore properties of this distribution further, however relatively little is known about it and it's behavior seemingly changes depending on the manifold.  Even the question of tail behavior can be unclear (e.g. the density function might not even be integrable).  Combined with Assumption 1, it is reasonable to clip the distribution outside of a ball if needed.  {Identification of such a ball will depend intimately on the sectional curvatures of the manifold under consideration}. We will add some discussion about the case of positive definite matrices where the Laplace distribution has been better studied as well as properties that would be interesting to explore.
>
> Thank you for pointing out your concern over Assumption 1, we intend to add a substantial amount of discussion concerning it when revising the paper. There are several important points you make that we want to clarify.  (1) Most DP methods require a similar assumption, namely, that the data is bounded in some way so as to control the global sensitivity.  {In our setting, since we work with the Fr\'echet mean, this translates to a radius requirement mainly for positively curved manifolds that ensures existence and uniqueness of the mean and has nothing to do with privacy. This, when coupled with having to consider the worst-case scenario when bounding the sensitivity, requires a convex (squared) distance function, which in turn further restricts the radius of the ball. Given these requirements, Assumption 1 represents the tightest set of sufficient conditions available in literature. In fact, this is true more generally for a $p$-Fr\'echet mean defined using the $p$th power of the distance function ($p=1$ results in the Fr\'echet median). Assumption 1 might be relaxed for other statistics on manifold different from the sample $p$-Fr\'echet mean.}
>
> (2) The proposal of projecting the data onto the tangent plane at $m_0$ is natural, but the mean on that tangent plane will not correspond to the Fr\'echet mean since the data will end up distorted.  Ideally, if one wanted to take such an approach, then we should project onto the tangent plane at the Fr\'echet mean.  However, the footpoint is then not private and one has a new set of problems; we originally explored this option, and there are some surprising challenges since one has to push the density from the tangent plane back to the manifold and prove DP there.
>
> (3) The determinant of the Jacobian of the exponential map is not constant; we will adjust the language to make this clearer.  All we require when proving our utility result is that it is bounded away from zero over $B_r(m_0)$, which follows from Assumption 1 (since the exponential map is diffeomorphic over that region).
>
> Lastly, our goal with the baseline comparison was just to highlight that incorporating the structure of the manifold will increase the performance of the original approach.  We felt we were even generous to the baseline approach since we used the Euclidean metric to evaluate performance.  We agree that depending on the manifold, there will be a variety of approaches from the literature one could use.
>
> We will clarify the minor comments (the reviewer is correct on all counts).  If we have the space, we can incorporate a graphic highlighting the various concepts from differential geometry.  If we are running low on space we will add a few sentences providing some additional intuition.

---

> > ### Comment · Reviewer_aauF · 2021-08-20
> > **Mean = Fr'echet mean**
> >
> > Thank you for your response, authors. I agree that after projecting from the tangent plane onto the manifold, the mean on the tangent plane will not correspond to the Fr'echet mean since the data will be distorted. However, my understanding of your comment that the mean of the Laplace distribution need not be the footprint implies that the same complaint could be leveled again the Laplace mechanism? Is that correct?

---

> > > ### Author Response · Authors · 2021-08-23
> > > **Asymptotic Bias**
> > >
> > > Any bias inherent in the Laplace mechanism is at least asymptotically negligible (as our utility result guarantees) the the mechanism will concentrate around the footpoint.  However, the same cannot be said about projecting to a single tangent plane, which will stay biased even asymptotically.  However, we agree that this is a subtle and interesting point that would be worth discussing after we introduce the Laplace mechanism.

---

### Decision · Program_Chairs · 2021-09-27

**Decision:**

Accept (Poster)

**Comment:**

This work studies the question of extending private mechanisms to the case when the output is known to lie on a manifold in R^d. A natural approach in such settings is to add noise as if the output was in R^d, and then project back to the manifold. The authors propose a different noise mechanism that lives on the manifold and can be potentially better in certain settings.
The reviewers had several useful comments that the authors will do well to address. Some of the discussion in the rebuttal helped clarify the paper's contribution and including some of that in the paper will help improve the paper.
I think the paper pushes the research in DP mechanisms forward in an potentially fruitful direction and I recommend accepting it.